# Artificial Intelligence in Detection, Management, and Prognosis of Bone Metastasis: A Systematic Review

**DOI:** 10.3390/cancers16152700

**Published:** 2024-07-29

**Authors:** Giuseppe Francesco Papalia, Paolo Brigato, Luisana Sisca, Girolamo Maltese, Eliodoro Faiella, Domiziana Santucci, Francesco Pantano, Bruno Vincenzi, Giuseppe Tonini, Rocco Papalia, Vincenzo Denaro

**Affiliations:** 1Operative Research Unit of Orthopaedic and Trauma Surgery, Fondazione Policlinico Universitario Campus Bio-Medico, Via Alvaro del Portillo, 200, 00128 Rome, Italy; g.papalia@policlinicocampus.it (G.F.P.);; 2Research Unit of Orthopaedic and Trauma Surgery, Department of Medicine and Surgery, Università Campus Bio-Medico di Roma, Via Alvaro del Portillo, 21, 00128 Rome, Italy; 3Department of Medical Oncology, Fondazione Policlinico Universitario Campus Bio-Medico, Via Alvaro del Portillo, 200, 00128 Rome, Italy; 4Department of Radiology and Interventional Radiology, Fondazione Policlinico Universitario Campus Bio-Medico, Via Alvaro del Portillo, 00128 Rome, Italy; 5Research Unit of Radiology and Interventional Radiology, Department of Medicine and Surgery, Università Campus Bio-Medico di Roma, Via Alvaro del Portillo, 21, 00128 Rome, Italy

**Keywords:** artificial intelligence, deep learning, machine learning, bone metastasis, orthopedic oncology

## Abstract

**Simple Summary:**

Bone metastases represent a serious and common challenge in advanced cancer patients, making early detection crucial. The growing implementation of artificial intelligence (AI) in various medical fields is revolutionizing healthcare by enhancing diagnostic accuracy. This systematic review includes 59 articles and sought to assess the potential of machine learning models in the fields of nuclear medicine, clinical research, radiology, and molecular biology. The review provides a comprehensive analysis of AI’s effectiveness in detecting bone metastases, which had not yet been thoroughly conducted. The findings highlight the benefits of integrating AI technologies in the inspected fields, encouraging further advancements and adoption of AI to ultimately improve patient care and treatment planning. This study provides invaluable perceptions into the promising application of computational intelligence in the detection of bone metastases, demonstrating its superiority over standard tests. However, further research is necessary to substantiate these findings with the aim of developing better-trained data-driven models soon.

**Abstract:**

Background: Metastasis commonly occur in the bone tissue. Artificial intelligence (AI) has become increasingly prevalent in the medical sector as support in decision-making, diagnosis, and treatment processes. The objective of this systematic review was to assess the reliability of AI systems in clinical, radiological, and pathological aspects of bone metastases. Methods: We included studies that evaluated the use of AI applications in patients affected by bone metastases. Two reviewers performed a digital search on 31 December 2023 on PubMed, Scopus, and Cochrane library and extracted authors, AI method, interest area, main modalities used, and main objectives from the included studies. Results: We included 59 studies that analyzed the contribution of computational intelligence in diagnosing or forecasting outcomes in patients with bone metastasis. Six studies were specific for spine metastasis. The study involved nuclear medicine (44.1%), clinical research (28.8%), radiology (20.4%), or molecular biology (6.8%). When a primary tumor was reported, prostate cancer was the most common, followed by lung, breast, and kidney. Conclusions: Appropriately trained AI models may be very useful in merging information to achieve an overall improved diagnostic accuracy and treatment for metastasis in the bone. Nevertheless, there are still concerns with the use of AI systems in medical settings. Ethical considerations and legal issues must be addressed to facilitate the safe and regulated adoption of AI technologies. The limitations of the study comprise a stronger emphasis on early detection rather than tumor management and prognosis as well as a high heterogeneity for type of tumor, AI technology and radiological techniques, pathology, or laboratory samples involved.

## 1. Introduction

Metastasis commonly occur in the bone tissue, making it one of the principal sites of invasion, notably for breast, prostate, and lung cancers [1]. The onset of bone metastases results in limitation of daily activities, pain, and reduced 5-year survival in cancer patients. Reports indicate that a poor prognosis is observed in 60–70% of patients with spinal metastases [2]. The typical survival time for individuals impacted by bone metastases from melanoma, breast, or prostate malignancy is 6 months, 19–25 months, and 53 months, respectively [3]. Furthermore, bone metastases are linked to a decline in the overall well-being, especially when hypercalcemia, uncontrolled pain, or skeletal-related events (SREs) occur [4]. These incidents characterize up to 70% of cases, particularly in patients in whom preventive interventions like the utilization of orthotic devices or bone-targeting agents (BTAs) are not implemented [5,6,7]. SREs include pathological fractures, spinal compression, radiation therapy, and metastasis surgery [8]. The rise in life expectancy among patients with bone metastases has led to a corresponding increase in SRE occurrences [9]. In the case of pathological fracture, surgical treatment aims to stabilize the fractured fragment [10], while in the presence of pathological vertebral collapse with spinal cord compression, urgent decompression surgery with possible stabilization of adjacent segments is required [11,12].

The diagnosis of bone metastases uses traditional radiology firstly, followed by second-level imaging techniques such as computed tomography (CT) and magnetic resonance imaging (MRI), and finally bone scintigraphy (BS) or positron emission tomography and computed tomography (PET-CT) [13].

Over the past few years, machine intelligence has proliferated in the medical domain as an aid in decision making, diagnosis, and treatment [14,15]. Already, in several fields, artificial intelligence (AI) applications such as convolutional neural networks (CNN), deep learning (DL), and machine learning (ML) have proven useful in determining predictive factors inherent in the pathology in question [16,17]. However, several limitations reduce the full integration of AI into clinical practice. In particular, AI systems for the treatment and diagnosis of tumors need to be implemented, and there are also concerns related to the availability and security of using big data [18,19]. Li et al. already analyzed the contribution of neural networks in the diagnosis of skeletal muscle oncological disorders [20]. 

However, a thorough examination of the efficacy and function of AI in bone metastasis remains unaccomplished. Hence, this study aims to evaluate the effectiveness of AI systems in the early detection, management, treatment, and evaluation of the prognosis of bone metastasis as a complementary tool useful in daily clinical practice. Therefore, this systematic review assesses the accuracy of the implementation of AI in nuclear medicine, clinical research, radiology, and molecular biology techniques for bone metastasis.

## 2. Materials and Methods

### 2.1. Inclusion Criteria

In this systematic review, studies that assessed the use of AI applications such as CNN, DL, or ML in patients with bone metastases were included. We excluded studies not written in English and animal and cadaveric studies. 

### 2.2. Information Sources and Search Strategy

Two reviewers (G.F.P. and G.M.) performed an electronic search on PubMed, Scopus, and Cochrane Library using the search strings (“deep learning” [MeSH Terms] OR (“deep” [All Fields] AND “learning” [All Fields]) OR “deep learning” [All Fields] OR (“machine learning” [MeSH Terms] OR (“machine” [All Fields] AND “learning” [All Fields]) OR “machine learning” [All Fields]) OR (“artificial intelligence” [MeSH Terms] OR (“artificial” [All Fields] AND “intelligence” [All Fields]) OR “artificial intelligence” [All Fields])) AND ((“bone and bones” [MeSH Terms] OR (“bone” [All Fields] AND “bones” [All Fields]) OR “bone and bones” [All Fields] OR “bone” [All Fields]) AND (“metastasi” [All Fields] OR “neoplasm metastasis” [MeSH Terms] OR (“neoplasm” [All Fields] AND “metastasis” [All Fields]) OR “neoplasm metastasis” [All Fields] OR “metastasis” [All Fields])). The research was conducted on 31 December 2023. Following the PRISMA 2020 statement [21], duplicates were removed from the identified records, and the two independent reviewers proceeded to screen the manuscripts respecting the inclusion and exclusion criteria. Finally, the fully eligible articles were read by the reviewers to select the studies to include in the review. 

### 2.3. Data Collection, Analysis, and Outcomes

Two reviewers (P.B. and G.M.) extracted these data from the included studies: authors, year, artificial intelligence method, interest area, main modalities used, and main objectives. The analysis included the assessment of data concerning the fields of nuclear medicine, clinical research, radiology, and molecular biology. This systematic review was not previously registered in databases.

## 3. Results

### 3.1. Search Results

The literary investigation identified 474 articles. After duplicate removal, 380 articles were screened based on title and abstract. The entire text of 77 papers was read, and 18 were excluded for the following reasons: not specific for bone metastases (*n* = 13) and not an assessment of AI (*n* = 5). The articles included in this review were 59 (Figure 1).

All the articles analyzed the role of AI in the diagnosis of skeletal metastasis or prognosis among individuals experiencing bone metastasis, of which six (10.2%) were specific for spine metastasis. The study involved nuclear medicine (*N* = 26 [44.1%]), clinical research (*N* = 17 [28.8%]), radiology (*N* = 12 [20.4%]), or molecular biology (*N* = 4 [6.8%]). When a primary tumor was reported, prostate neoplasms were the prevailing cancer (*N* = 13 [22%]), preceding lung (*N* = 7 [11.9%]), breast (*N* = 3 [5.1%]), kidney (*N* = 3 [5.1%]), thyroid (*N* = 1 [1.7%]), and colorectal cancer (*N* = 1 [1.7%]). 

### 3.2. Nuclear Medicine

Imaging modalities in nuclear medicine represent a fundamental tool in the diagnosis and management of metastatic bone disease [22]. The use of AI in nuclear medicine for bone metastasis is evolving quickly, showing new opportunities for diagnosis, treatment, and patient care. AI can improve the analysis of nuclear medicine imaging techniques by offering detailed quantitative evaluations of tracer uptake and delivering accurate localization of bone metastases. In our research, nuclear medicine was the most common category (*N* = 26). 

#### 3.2.1. Bone Scintigraphy

The main modality used was bone scintigraphy (*N* = 21 [80.77%]). Research demonstrated that machine learning algorithms utilizing BS images can accurately differentiate the metastatic bone tissue from normal tissue [13]. Zhao et al. [23] demonstrated a significant diagnostic performance of ML techniques based on Tc-MDP BS in the diagnosis of skeletal metastasis, showing an area under the curve (AUC) of receiver operating characteristic (ROC) of 0.988 for breast cancer, 0.955 for prostate cancer, 0.957 for lung cancer, and 0.971 for other cancers. Groot et al. [24] developed an NLP algorithm designed to classify single or multiple metastases in bone scans of individuals chosen for surgical treatment of bone metastases, which had a sensitivity of 0.94 and specificity of 0.82. 

Koizumi et al. [25] evaluated the performance of the BONEVAVI version 2 for bone metastasis diagnosis among individuals having or lacking skeletal metastasis. The study reported high levels of sensitivity and specificity of patient artificial neural network (ANN) data, reaching values higher than 80%. 

Koizumi et al. [26] also investigated the diagnostic performance of the computer-assisted diagnostic system for BS BONENAVI in the presence or absence of bone involvement in prostate cancer, assessing an accuracy of 82% for metastasis detection. Papandrianos et al. [27] used a CNN algorithm for BS in determining the occurrence or non-occurrence of prostate cancer metastasis. They demonstrated that the method is adequately accurate in distinguishing between bone metastasis and degenerative or normal tissues, achieving an overall classification accuracy of 91.42% ± 1.64%. Finally, the best-performing CNN method outperformed commonly used CNN methods in nuclear medicine for diagnosing metastatic prostate cancer in bones [28]. 

#### 3.2.2. Single-Photon Emission-Computed Tomography

In our study, single-photon emission-computed tomography (SPECT) ranked as the second most frequently utilized modality (*N* = 5 [19.23%]). It was shown that deep learning models can accurately identify hotspots of metastasis in bone SPECT images, achieving values of 0.9920, 0.7721, and 0.6788 for accuracy, precision, and recall, respectively [29]. Moreover, Lin et al. presented similar results in other manuscripts [30,31]. Acar et al. [32] used ML algorithms based on 68Ga-prostate-specific membrane antigen (PSMA) PET-CT images to distinguish sclerotic lesions from metastasis or completely responded lesions in individuals with acknowledged bone metastasis who had undergone prior treatment.

### 3.3. Clinical Research

AI models for clinical prediction have been established for many purposes using clinical data obtained from medical records. AI can examine large clinical datasets to obtain an early diagnosis, create individualized treatment plans, and deliver customized prognostic insights for each patient. De Groot et al. [33] analyzed ML algorithms in predicting the 90-day and 1-year survival in subjects with skeletal metastasis who had surgery. Zhou et al. analyzed the diagnostic performance of ML algorithms in patients with lung adenocarcinoma bone metastasis [34]. In both the training and test groups, the AUC values of all ML classifiers exceeded 0.8, except for random forest (RF) and logistic regression (LR). However, the combined algorithm did not enhance the AUC value for any individual machine learning algorithm. The accuracy of all ML classifiers, except for the RF algorithm, surpassed 70%. ML algorithms were used also in predicting outcomes for cancer-specific survival (CSS) and overall survival (OS). For example, Chen et al. [35] found that patients with small-cell lung cancer (SCLC) with skeletal metastasis experienced a reduced median survival time (MTS) for OS compared to patients with SCLC without bone metastasis (6 vs. 10 months). 

Le et al. [36] investigated the function of ML algorithms in forecasting the 3-year OS of patients diagnosed with bone metastasis from clear cell renal cell carcinoma, proving that the extreme gradient boosting (XGB) algorithm model achieved the best accuracy and specificity (0.792 and 0.806, respectively) compared to other models. Finally, Paulino Pereira et al. [37] analyzed the performance of ML algorithms in estimating survival in patients surgically treated for spinal metastases.

### 3.4. Radiology

In our review, radiology-related articles involved CT and MRI. CT (*N* = 8 [66.7%]) was the most common imaging modality used. 

#### 3.4.1. Computed Tomography

AI applications in CT for bone metastasis provide substantial potential benefits such as great diagnostic accuracy, efficient segmentation of the lesions (Appendix A) and treatment planning, and patient follow-up by identifying early signs of disease progression or treatment response. In the study by Noguchi et al. [38] to evaluate efficiency in skeletal metastases detection on CT, in the validation dataset, the deep learning-based algorithm attained a sensitivity of 89.8% with 0.775 false positives per case. 

#### 3.4.2. Magnetic Resonance Imaging

The second most common modality used was MRI imaging (*N* = 4 [33.3%]), of which one was pelvic MRI, and three were whole-body MRI. AI algorithms can greatly improve the sensitivity and specificity of MRI scans for detecting bone metastases, identifying subtle changes in bone marrow and other structures that human radiologists might overlook. Additionally, AI can automate MRI image analysis, which can reduce the time needed for diagnosis and result in faster and more efficient clinical workflows. Tajima et al. [39] assessed the image acquisition time in 17 patients with prostate malignancy who had undergone diffusion-weighted whole-body imaging with background body signal suppression by 1.5T MRI with 2 excitations (NEX2) and 8 (NEX8). 

AI can be also used in diagnosis of spine metastasis. In particular, Wang et al. [40] analyzed the CNN algorithm based on MRI spine scans to diagnose and predict spinal metastasis. They found that this model could accurately forecast spinal metastases, with an accuracy up to 96.45%. Furthermore, Jakubicek et al. [41] examined how ML techniques utilizing vertebrae in 3D CT scans perform in diagnosing potentially incomplete spines in patients with bone metastases and vertebral compressions, showing a mean error of intervertebral discs localization of 4.4 mm.

### 3.5. Molecular Biology

AI can process extensive datasets from genomics, proteomics, and metabolomics to reveal the molecular mechanisms associated with bone metastasis. This could enhance the understanding of the disease and help identify potential therapeutic targets. Additionally, AI can spot molecular biomarkers indicative of bone metastasis, facilitating early detection and personalized treatment strategies. Shao et al. [42] used label-free surface-enhanced Raman spectroscopy for screening prostate carcinoma bone metastasis, finding high values for mean training accuracy (99.51%), testing accuracy (81.70%), testing sensitivity (80.63%), and testing specificity (82.82%). Albaradei et al. [43] used bone metastasis-related genes from gene expression datasets in Gene Expression Omnibus (GEO) to predict bone metastases development. The deep neural network (DNN) model achieved the peak prediction accuracy (AUC of 92.11%) by utilizing the top 34 genes identified based on their betweenness centrality scores. Hsu et al. [17] analyzed patients who underwent local MRI-guided focused ultrasound ablation to determine an appropriate treatment plan for skeletal metastasis. The most accurate predictive few-shot learning model was obtained by integrating clinical features with the amount of cytokines IL-6, IL-13, IP-10, and eotaxin (accuracy of 85.2% and sensitivity of 88.6%). Park et al. [44] examined breast cancer genes frequently expressed during bone metastasis and in osteoblasts using data from the GEO database to create a potential causal network. They discovered 33 genes strongly linked to the onset of breast cancer skeletal metastasis. Additional model assessments revealed that 16 genes were sufficient to make the model statistically significant for the maximum likelihood of causal Bayesian networks and accurately predict breast cancer bone metastasis.

All the included studies with their characteristics are reported in Table 1.

## 4. Discussion

AI has gained great importance in the medical field over the last 10 years, representing a breakthrough in the scientific literature [78]. AI use has been primarily confined to evaluating bone metastasis using bone scintigraphy and analyzing histopathology samples in the diagnosis of both solid and hematological neoplasms [79]. Electronic health records or molecular datasets are still relatively unknown fields for AI models. The application and continued usage of these models may revolutionize these domains, making them a more powerful tool than those available at this moment [80]. The field of musculoskeletal oncology may benefit from a mix of discoveries both from clinical and non-clinical specialties [81]. Nuclear medicine has long played a significant role in musculoskeletal oncology, with numerous new imaging techniques investigated over time [82], such as PET/CT and multiparameter MRI. Despite this, bone scintigraphy with 9mTc-MDP continue to have a noteworthy position thanks to the advantages of being less expensive and still having decent sensitivity [83]. As shown by Zhao et al. [23], AI models with deep neural networks using image features from bone scintigraphy with 9mTc-MDP demonstrated notable time efficiency, accuracy, specificity, and sensitivity in diagnosing bone metastasis, making it potentially valuable in distant or economically disadvantaged areas where interpreting bone scintigraphy images remains challenging.

Single-photon emission-computed tomography is also a widely used technique of nuclear medicine, which can evaluate the metabolism of radioactively labeled compounds, bringing useful information about a specified area of concern that may present itself with malignancy features, also referred to as hotspots [84]. Deep learning-based segmentation models using SPECT images showed promising results in detecting and segmenting hotspots [85], with propitious future directions including the necessity of optimization and improvement of the models to develop a more straightforward, efficient, and productive computer-assisted diagnosis system and to evolve into multi-category and multi-condition models for identifying hotspots of different diseases in SPECT thorax images [29].

Computed tomography is a primary imaging technique employed in diagnosing a wide range of illnesses. In particular, even though it is considered inferior to MRI and PET for detecting bone metastases, especially those affecting the spine, its widespread adoption allows for routinely performed exams, making it an ideal candidate for computer-aided detection (CAD) systems [86] under AI, including ML and DL [87,88]. For all the above-mentioned reasons, Koike et al. [75] created a fully automated two-step CAD method with AI assistance for identifying lytic spinal bone metastases and showed promising results, as the system was both accurate and rapid compared to the conventional diagnosis of spinal metastasis, which usually comes with the possibility of overlooking the lesions if the examinations are not specifically designed to assess them. On the other hand, MRI offers an overall accuracy of 70–100% in identifying bony metastases [89]. It is broadly used in the diagnosis and staging of bone lesions. The combination of MRI and PET/CT parameters has been shown to have the possibility of predicting early metastatic disease if analyzed by AI models such as model-averaged neural network (avNNet) [90]. However, all the above-mentioned radiological and nuclear medicine techniques are associated with high costs and radioactive injury, which are variables not to be ignored. For this reason, low-cost, rapid, and accurate laboratory tests that specifically evaluate alterations in bone metabolism in the presence of bone metastasis have sparked considerable interest. For example, increased levels of procollagen N-terminal propeptides (PIPNs), osteocalcin, and bone-specific-alkaline phosphatase (BAP) could indicate heightened osteoblastic activity, which may represent an underlying metastatic condition of the patient [91]. Yet, the assessment of bone metabolic parameters is not currently advised for routine clinical practice due to conflicting findings. For this reason, Raman spectroscopy (SERS) has seen increased usage over the years as a method for blood analysis thanks to the relatively fast results and the possibility of predicting disease recurrence based on serum parameters, for example, in the case of the prostate [92,93]. Shao et al. [42] implemented Raman spectra parameters on prostate screening tests for convolutional neural networks (CNNs) to identify early disease, with promising results. Raman spectroscopy was also used in the evaluation of breast carcinoma genes frequently expressed in individuals with bone metastasis to construct a plausible causal network. In the study by Park et al. [44], 33 genes that were significantly associated and likely involved in the progression of breast cancer bone metastasis were identified. Among these, 16 genes were sufficient for a model to accurately predict breast cancer bone metastasis. In conclusion, clinical-prediction AI models have been developed over the years with many applications, one of which is the evaluation of disease-free survival (DFS) and overall survival (OS) in cancer patients [2]. Regarding bone metastasis, survival after the diagnosis is quite variable and has an essential role in care decision making. For example, the prophylactic stabilization of impending pathologic fractures in the case of bone metastasis is heavily influenced by the predicted survival of the patient, with a relative contraindication in patients with 3 months or less of estimated survival [94]. For this reason, many predictive models for the survival of patients with various diseases and at different stages and for candidates for various medical procedures proposed over time span from traditional statistics to boosting algorithms and, more recently, machine learning models [95,96]. The latter has the advantage of being able to delineate predictive models based on larger datasets. In addition to this, AI models are subjected to retraining because their predictive performance tends to decline over time as treatment regimens tend to advance. As the complexity of these networks increases, AI-based models require significant computational power and memory. Moreover, such complex networks may not always lead to faster performance, especially when implemented on devices with limited resources. In medical cases, time is crucial for early diagnosis of bone metastasis to improve the clinical outcomes of the patients with bone metastasis using the best treatment choices [97]. Nowadays, the greatest challenge is identifying how to improve the AI-based model systems used in medical applications in terms of feasibility, reproducibility, and fastness in processing data. AI applications are increasing dramatically, with promising but intricate prospects. Regarding the early detection of metastasis, enhancing detection accuracy may be crucial [98]. This will be driven by expanding datasets, advancing machine learning algorithms, and improving imaging technologies [99]. Concurrently, advanced imaging algorithms may aid in distinguishing metastasis from other pathologies, enhancing the resolution and clarity of medical images, and shortening the diagnostic phase, thereby expediting the initiation of patient treatment [100]. Furthermore, integrating established AI models may help reduce human error during image evaluation. The clinical management and prognosis of tumor metastasis may also be shaped by the deployment of well-trained AI models. For instance, the integration of genetic, demographic, and clinical data could lead to personalized treatment plans becoming standard practice, where treatments are tailored to individual patients rather than generalized. This integration offers a holistic perspective on a patient’s health, which aids in making informed decisions and coordinating care more effectively [101]. Moreover, by integrating patient data, it becomes possible to predict the progression of bone metastasis, assisting in the formulation of long-term treatment strategies and the management of disease progression [102]. Nevertheless, there are still some concerns with the use of AI systems in medical settings. For instance, it is essential to ensure that the diverse datasets on which AI systems are trained can precisely represent different patient populations, that they are effectively integrated into routine clinical workflows, and that they can be improved in overall interpretability and transparency. Furthermore, ethical considerations and regulatory and legal issues must be addressed to facilitate the safe and regulated adoption of AI technologies in bone metastasis detection and overall management [103]. There are some limitations to this study. The majority of studies included focused on early detection of bone metastasis, and only a few focused on tumor management and the treatment and evaluation of prognosis. Hence, future studies that evaluate the clinical management of metastasis rather than the radiological features must be considered. Furthermore, the included studies presented a high grade of heterogeneity regarding the type of tumor evaluated and both the AI technology and the radiological techniques, pathology, or laboratory samples implemented.

## 5. Conclusions

The diagnosis and management of tumors in the musculoskeletal system are extremely complex, requiring a multidisciplinary approach. In this context, appropriately trained AI models hold significant promise for enhancing the precision of diagnosing and treating bone metastasis. By effectively merging and analyzing data from radiologic imaging, pathology, and laboratory samples, AI can provide a comprehensive and nuanced understanding of tumor characteristics. This can lead to more accurate and timely diagnoses, personalized treatment plans, and improved patient outcomes. Furthermore, the continuous advancement of AI technologies and their integration into clinical workflows could revolutionize the standard of care in musculoskeletal oncology, paving the way for more effective and efficient management of bone metastasis. Future research should focus on the development of robust and reliable AI algorithms, validation of these algorithms in diverse clinical settings, and addressing potential ethical and implementation challenges to fully realize the potential of AI in the field.

## Figures and Tables

**Figure 1 cancers-16-02700-f001:**
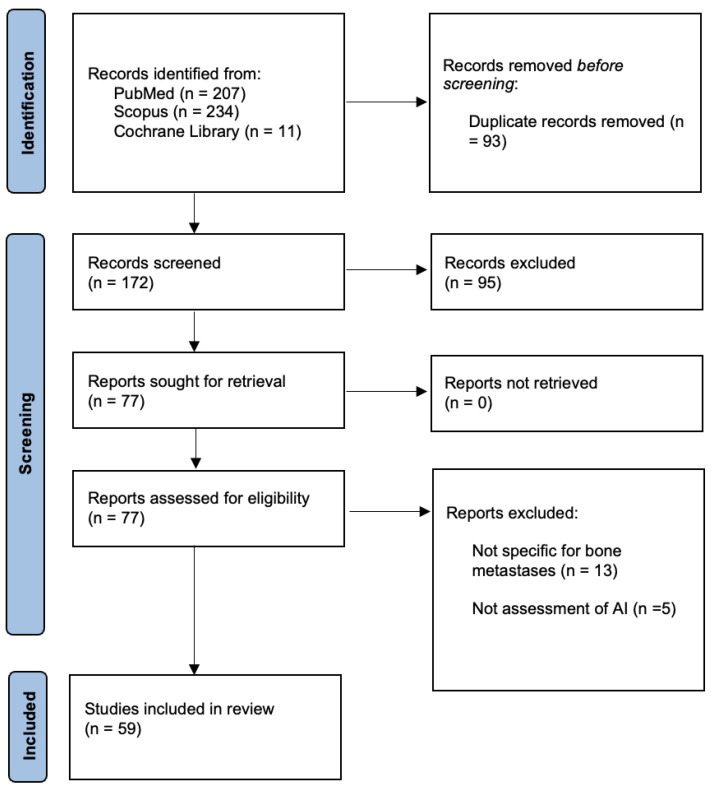
PRISMA 2020 flow diagram.

**Table 1 cancers-16-02700-t001:** AI methods and results of included studies.

Authors	AI Method	Interest Area	Main Modality Used	Main Objectives	Results
Elfarra et al. (2019) [13]	PC	Nuclear Medicine	BS	Diagnosis of metastatic tissue on BS.	The method was able to discriminate skeletal metastasis from normal tissue (accuracy = 87.58%).
Koizumi et al. (2017) [26]	ANN	Nuclear Medicine	BS	Diagnostic performance of CAD system for BS BONENAVI in prostate cancers in the presence or absence of bone metastasis.	BONENAVI obtained sensitivity = 82% and specificity = 83% in detecting metastases. No difference was found based on CT types.
Koizumi et al. (2020) [45]	ANN	Nuclear Medicine	BS	Effectiveness of CAD system for BS BONENAVI as diagnostic tool for bone metastasis.	ANN sensitivity = 76%. ANN values were related to BS type.
Zhao et al. (2020) [23]	DNN	Nuclear Medicine	BS	Performance of Tc-MDP BS as diagnostic tool for of skeletal metastasis.	AUC of ROC = 0.988 for breast cancer, 0.955 for prostate cancer, 0.957 for lung cancer, and 0.971 for others.
Lin et al. (2021) [46]	CNN	Nuclear Medicine	Thoracic SPECT	Automatically diagnosing bone metastasis in thoracic SPECT images.	Tests on SPECT bone images proved that the proposed classifiers were effective in detecting skeletal metastases with SPECT imaging (accuracy = 0.9807, precision = 0.9900, specificity = 0.9890, and AUC = 0.9933).
Papandrianos al. (2020) [47]	CNN	Nuclear Medicine	BS	Performance of an algorithm for prostate cancers to detect bone metastasis.	The method was able to discriminate skeletal metastasis from degenerative changes or normal tissue (accuracy = 91.42%).
Aoki et al. (2020) [48]	DL	Nuclear Medicine	BS	Utility of DL-based algorithm for prostate cancers to determine the presence of skeletal metastases.	No significant differences were shown between the per-patient detection rates performed by the specialists and the software.
Acar et al. (2019) [32]	DT, SVM, KNN, discriminant analysis, ensemble classifier	Nuclear Medicine	68Ga-PSMA PET/CT	Recognizing the lesions obtained by 68Ga-PSMA PET/CT as metastases in presence of known skeletal metastases.	The weighted KNN model obtained the best accuracy. This algorithm successfully distinguished sclerotic from metastatic lesions with AUC = 0.76.
Cheng et al. (2021) [49]	CNN, DL	Nuclear Medicine	BS	Early diagnosis of skeletal metastasis.	For detecting and classifying skeletal metastasis in the chest of prostate patients on a lesion basis, the average sensitivity was 0.72, and the average precision was 0.90. For classifying skeletal metastasis on a patient basis, the average sensitivity was 0.94, and the average specificity was 0.92.
Chiu et al. (2009) [50]	ANN	Nuclear Medicine	Tc-99m MDP whole-body BS	ANN model as tool in bone metastasis for predicting in prostate cancers.	AUC under the ROC curve showed the highest simultaneous sensitivity (87.5%) and specificity (83.3%).
Liao et al. (2023) [51]	(Fast and faster) region-based CNN with RPN	Nuclear Medicine	99mTC-labeled bisphosphonates	Increasing the effectiveness of detecting skeletal metastasis detection on bone scans.	Optimal DSC = 0.6640, differing by 0.04 relative to optimal DSC of physicians (0.7040).
Lin et al. (2022) [30]	RPN	Nuclear Medicine	99mTc-MDP SPECT	Developing a deep image segmentation model that automatically identifies and delineates lesions of skeletal metastasis in bone scan images.	The proposed algorithm achieved good results for automatic segmentation of metastatic lesions, with a DSC score of 0.692.
Groot et al. (2020) [24]	NLP	Nuclear Medicine	BS	NLP algorithm to distinguish single metastasis from ≥2 metastases in BS of patients who underwent surgical treatment.	With a threshold of 0.90, NLP model accurately detected multiple skeletal metastases in 117 out of 124 cases (sensitivity = 0.94) and produced three false positives (specificity = 0.82).
Ntakolia et al. (2020) [52]	CNN	Nuclear Medicine	BS	Evaluating the performance of the CNN in bone metastasis detection.	Higher performance in detecting skeletal metastasis of the proposed method compared to the actual methods.
Inaki et al. (2019) [53]	ANN	Nuclear Medicine	BS	Effectiveness of ANN-based quantitative BS in the diagnosis in breast cancers.	Extent of disease, BSI, SUVmax, TLG, MTV, and serum tumor markers were significantly higher in patients with skeletal metastases compared to those with no metastases. In multivariate Cox proportional hazard model, BSI and SUVmax represented prognostic factors for patients without visceral metastases.
Kikushima et al. (2015) [54]	ANN	Nuclear Medicine	BS	Evaluating the accuracy of CAD system for BS BONENAVI version 2, in patients with suspected skeletal metastasis.	BN2-Sp and BN2-Sen showed similar effectiveness to BN2-B in identifying patients with skeletal metastases. Overall, 65.4% of patients presented concordance for “bone metastases” or “no bone metastases”, while 34.6% presented a mismatch.
Koizumi et al. (2015) [25]	ANN	Nuclear Medicine	BS	Evaluating the diagnostic performance of the BONEVAVI version 2 in presence or absence of skeletal metastasis.	ANN sensitivity was 85% for all cancers, 86% for prostate, 88% for lung, 82% for breast, and 86% for others. ANN specificity was 82% for normal bone scans, 81% for consecutive patients with several days of no bone metastasis, and 54% for abnormalities at bone scans without bone metastasis.
Lin et al. (2020) [29]	DL	Nuclear Medicine	Thoracic bone SPECT	Automatic delineation of hotspots boundaries in skeletal SPECT images based on DL segmentation models.	The segmentation models were able to identify and segment hotspots of metastases in bone SEPCT images, reaching accuracy = 0.9920 and precision = 0.7721.
Lin et al. (2022) [31]	DL	Nuclear Medicine	SPECT	Automatic identification and localization of hotspots in bone scans with lung cancer metastatic lesions.	Tests on clinical data of retrospective bone scans presented similar performance with precision = 0.7911. A comparative analysis demonstrated that automatic detection of multiple lung cancer metastatic lesions is feasible.
Liu et al. (2021) [55]	CNN	Nuclear Medicine	Whole-body BS	CNN-based detection of suspect skeletal metastases from whole-body BS	The proposed network obtained the highest accuracy (81.23%) in the detection of suspected skeletal lesions. The CNN model’s lesion-based mean sensitivity was 81.30%, and mean specificity was 81.14%.
Liu et al. (2022) [56]	DL	Nuclear Medicine	BS	DL-based automatic analysis of bone metastasis on BS.	The classification model showed sensitivity = 92.59%, specificity = 85.51%, and accuracy = 88.62% in testing set. A positive correlation was reported between BSI and ALP level.
Papandrianos et al. (2020) [27]	CNN	Nuclear Medicine	BS	Evaluating the performance of CNN as diagnostic tool for skeletal metastases in prostate cancers.	The method demonstrated high precision in distinguishing skeletal metastases from degenerative modifications or normal tissue, with accuracy = 91.61%.
Pi et al. (2020) [57]	CNN	Nuclear Medicine	BS	Determining the absence or presence of bone metastasis.	High accuracy was demonstrated for the diagnosis with BS scans.
Papandrianos et al. (2020) [28]	CNN	Nuclear Medicine	BS	Evaluating CNN models that classifies BS scans, distinguishing between with or without prostate cancer metastasis.	The proposed CNN-based method was better than popular nuclear medicine CNN approaches in diagnosis of skeletal metastasis from prostate cancer.
Higashiyama et al. (2021) [58]	CNN	Nuclear Medicine	BS	Effectiveness of BSI calculated by CNNapis in prostate cancers skeletal metastasis.	Diagnosis of bone metastasis on BS was confirmed with CNNapis. A positive correlation was found between PSA and BSIm and ALP and BSI.
Yu et al. (2021) [59]	IFV	Nuclear Medicine	Tc-99m MDP whole-body BS	Self-developing IFV tool in prediction of suspected skeletal lesions in whole-body BS.	IFV model obtained sensitivity = 93% for prostate cancer, 91% for breast cancer, and 83% for lung cancer, showing better accuracy than BONEVAVI.
Thio et al. (2019) [16]	SORG, RF, SVM, neural network, penalized LR	Clinical Research		Assessing 90-day and 1-year survivals of patients undergoing surgery for extremity bone metastasis.	There was no significant difference in discrimination between the 5 models. Low levels of albumin and rapid growth of the primary tumor were associated with poorer 90-day and 1-year survival.
Zhou et al. (2023) [34]	LR, RF, GBM, DT	Clinical Research		Identifying lung adenocarcinoma. skeletal metastases.	The algorithm did not show improvement in AUC for any single ML algorithm both in training and in test group.
Anderson et al. (2022) [60]	GBM modeling	Clinical Research		Estimating overall survival after SREs treatment in men with prostate cancer bone disease metastases.	Young age at metastasis diagnosis, proximal PSA < 10ng/mL, and slow-rising APV were associated with higher survival.
Liu et al. (2020) [61]	DT, RF, MLP, LR, NBC, XGB	Clinical Research		Predicting skeletal metastases in thyroid cancer of new diagnosis.	RF model showed higher predictive accuracy compared to other models.
Liu et al. (2021) [62]	DT, RF, MLP, LR, NBC, XGB	Clinical Research		Predicting prostate cancers bone metastasis.	XGB model presented the highest predictive accuracy of the 6 models.
Xu et al. (2022) [14]	LR, RF, DT, GBM, XGB, NBC	Clinical Research		Estimating the risk of renal cell carcinomas skeletal metastases.	XGB model reported the highest prediction accuracy among the risk prediction models.
de Groot et al. (2022) [33]	SORG, RM, SVM, neural network, penalized LR	Clinical Research		Predicting the 90-day and 1-year mortality of patients who undergo surgery for skeletal metastasis	The AUC was 0.78 for 90-day survival and 0.75 for 1-year survival.
Xiong et al. (2022) [63]	LR, gradient boosting tree model, DT, RF	Clinical Research		Assessing the risk of early mortality in patients with skeletal metastasis from breast cancer.	GBM had the highest AUC (0.829), followed by RF and LR.
Li et al. (2023) [64]	LR, DT, RF, GBM, NBC, XGB	Clinical Research		Predicting non-small-cell lung cancers skeletal metastases.	Of the six models, the ML model built by the XGB algorithm performed best in internal and external data setting validation.
Cui et al. (2022) [65]	RF, GBM, DT, XGB	Clinical Research		Predicting 3-month survival of bone metastasis patients with unknown primary tumor.	The RF algorithm obtained the highest AUC value (0.796) and the second-highest precision (0.876) and accuracy (0.778).
Chen et al. (2023) [35]	DT	Clinical Research		Predicting outcomes for CSS and OS in patients with SCLC bone metastasis.	Patients with SCLC with bone metastasis had a reduced MST compared to those without bone metastasis, with significant Kaplan–Meier analysis (*p* < 0.05).
Li et al. (2022) [66]	LR, RF, SVM, DT, XGB	Clinical Research		Predicting probability of developing bone metastasis in colorectal cancer patients.	SVM algorithm with kernel function showed the best performance. The most important predictors of skeletal metastasis were extraosseous metastases, size, and CEA.
Ji et al. (2022) [67]	LR, NBC, DT, XGB, MLP, RF, SVM, KNN	Clinical Research		Evaluating risks and prognosis of skeletal metastases from kidney cancer.	The prognosis model achieved an AUC of 0.8269 in internal and 0.9123 in external validation cohort.
Cui et al. (2022) [68]	LR, XGB, RF, neural network, GBM, DT	Clinical Research		Estimating 3-month survival of patients with skeletal metastasis from lung cancer.	The GBM model outperformed all the other models, followed by XGB and LR. Important predictors in the population were chemotherapy, followed by liver metastases, radiation, and brain metastases.
Paulino Pereira et al. (2016) [37]	Boosting algorithm	Clinical Research		Estimating mortality in patients with spine metastasis who underwent surgery.	The boosting algorithm showed the highest survival prediction on training datasets.
Le et al. (2023) [36]	XGB, LR, RF, NBC	Clinical Research		Predicting the OS of patients presenting bone metastasis from clear cell renal cell carcinoma.	Compared to the other three models, XGB model had the highest accuracy, specificity, and F1 score in the prediction of the 1-year OS.
Jacobson et al. (2022) [69]	XGB	Clinical Research		Identifying patients with skeletal metastases from solid tumor with increased risk of SREs after Denosumab interruption.	The model identified significant factors for the prediction of higher SREs risk after Denosumab interruption as previous SREs, short treatment with Denosumab, young age at skeletal metastases, and prostate cancer.
Tajima et al. (2022) [39]	DL-based reconstruction method	Radiology	Whole-body MRI	Assessing a time reduction for image acquisition for DWIBS by denoising with reconstruction based on DL in prostate cancer patients.	NEX2 acquisition time was 2.8 times shorter than NEX8. No significant differences were found between dDLR-NEX2 and NEX8 in qualitative analysis.
Fan et al. (2021) [70]	Chan–Vese algorithm and AdaBoost algorithm	Radiology	MRI	Assessing the early detection on imaging of spine metastasis from lung cancer.	Jaccard coefficient and Dice index of Chan–Vese algorithm were better than region-growing algorithm and OTSU.
Noguchi et al. (2022) [38]	DL	Radiology	CT	Developing and evaluating an algorithm based on DL for automatic detection of skeletal metastasis on CT.	The DL-based algorithm reported sensitivity = 89.8% for the validation dataset and 82.7% for the test dataset.
Han et al. (2022) [71]	CNN	Radiology	CT	Evaluating the performance of DL in classifying bone scans of prostate cancers.	The ROC curves showed excellent performance of diagnosis for the AUC.
Huo et al. (2023) [72]	Deep CNN	Radiology	CT	Developing and assessing a deep CNN model for automatic CT assessment of lung cancer skeletal metastases.	The deep CNN model showed detection sensitivity = 0.894 and segmentation dice coefficient = 0.856.
Hong et al. (2021) [73]	RF	Radiology	CT	Evaluating the CT radiomics-based ML model performance in diagnosis and differentiation of bone islands from osteoblastic skeletal metastasis.	Mean AUC = 0.89 for the RF model; mean AUC = 0.96 for the trained RF model.
Hoshiai et al. (2022) [74]	DL	Radiology	CT	Assessing clinical performance of CT temporal subtraction with DL in improving the detection of spinal metastases.	Temporal subtraction CT was efficient in the detection of skeletal metastases.
Koike et al. (2023) [75]	DL	Radiology	CT	Detecting and classifying lytic spine metastases on CT scans.	AI-aided CAD system was able to recognize lytic vertebral metastases on CT scans.
Jakubicek et al. (2019) [41]	CNN	Radiology	CT	Vertebral detection on 3D CT scans of spinal metastasis and spinal cord compression.	The mean rate of correctly detected vertebral level was 87.1%.
Jin et al. (2023) [76]	ANN, RF, DT, SVM	Radiology	Pelvic multiparameter MRI	Building a GLCM-based score prediction model for skeletal metastases.	DWI DL-based model showed high accuracy in the automatic segmentation of pelvic bone, allowing the radiomics model to identify metastasis in the pelvis and evaluating pelvic bone turnover of colorectal cancer patients.
Masoudi et al. (2021) [77]	DL	Radiology	CT	Formulating an efficient DL-based classification method for CT scan characterization of skeletal metastasis from prostate cancer.	Accuracy = 92.2% in distinction of benign vs. malignant skeletal lesions based on texture, volume, and morphology of the lesions.
Wang et al. (2023) [40]	CNN	Radiology	MRI	Diagnosing and predicting spinal metastasis.	Accuracy of 96.45% in prediction of spine metastasis.
Shao et al. (2020) [42]	CNN	Molecular Biology	LF SERS	Screening of skeletal metastasis from prostate cancer.	CNN model showed for skeletal metastasis detection: training accuracy = 99.51%, testing accuracy = 81.70%, testing sensitivity = 80.63%, and testing specificity = 82.82%.
Albaradei et al. (2021) [43]	SVM, RF, DNN	Molecular Biology	Bone metastasis-related genes from gene expression datasets in GEO	Predicting bone metastases development.	DNN model showed the highest prediction accuracy (AUC = 92.11%) utilizing the top 34 ranked genes.
Hsu et al. (2022) [17]	RF, generalized linear, SVM, naive Bayesian models	Molecular Biology	Eotaxin and cytokines IL-6, IL-13, and IP-10	Providing guidelines for clinicians to determine an appropriate treatment plan for bone metastases.	The combination of clinical characteristics and levels of eotaxin or cytokines IL-6, IL-13, and IP-10 was the most effective predictive learning model (accuracy = 85.2%, sensitivity = 88.6%, and AUC = 0.95).
Park et al. (2018) [44]	CBN	Molecular Biology	Breast cancer genes expressed during bone metastasis and in osteoblasts from the GEO	Obtaining a network of gene interactions involved in bone metastasis and osteoblast activity in breast cancer.	33 related and involved genes (as levels of HEBP1, UBIAD1, TSPO, BTNL8, ZFP36L2, and PSAT1) in the onset of breast cancer bone metastasis were identified.

AI: artificial intelligence; BS: bone scintigraphy; PC: parallelepiped classification; PET: positron emission tomography; PSMA: prostate-specific membrane antigen; ANN: artificial neural network; DL: deep learning; CAD: computer-assisted diagnostic; DNN: deep neural network; Tc-MDP: Technetium-99m methylene diphosphonate; AUC: area under the curve; ROC: receiver operating characteristic; CNN: convolutional neural network; SPECT: single-photon emission-computed tomography; SVM: support vector machine; BSI: Bone Scan Index; KNN: k-nearest neighbor; SUV: standardized uptake value; MTV: metabolic tumor volume; TLG: total lesion glycolysis; IFV: irregular flux viewer; DT: decision tree; RPN: region proposal network; GBM: gradient boosting machine; MLP: multilayer perceptron; LR: logistic regression; NBC: naive Bayes classifiers; XGB: eXtreme gradient boosting; CSS: cancer-specific survival; OS: overall survival; SORG: stochastic gradient boosting; DSC: Dice similarity coefficient; PSA: prostate-specific antigen; APV: alkaline phosphatase velocity; SCLC: small-cell lung cancer; MST: median survival time; SRE: skeletal-related event; MRI: magnetic resonance imaging; GLCM: gray-level co-occurrence matrix; DWI: diffusion-weighted imaging; DWIBS: diffusion-weighted whole-body imaging with background body signal suppression; LF: label-free; SERS: surface-enhanced Raman scattering; RF: random forest; GEO: Gene Expression Omnibus; CBN: causal Bayesian networks.

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
