# Peer review of "Artificial Intelligence in Detection, Management, and Prognosis of Bone Metastasis: A Systematic Review"

_cancers, 2024, doi:10.3390/cancers16152700_

Round 1
Reviewer 1 Report
Comments and Suggestions for Authors
The article entitled “Artificial intelligence and bone metastasis: a systematic review” is well-written and, from my point of view, would be of interest for the readers of Cancers. In spite of this and before its publication, I would like to make the authors the following suggestions:
Introduction section: a summary of the aim of the manuscript and, also, a description of the layout of the manuscript should be introduced.
Lines 86-88 please rewrite the sentence in order to make it easier to understand.
As a suggestion, search strings would be presented in a table.
Figure 1: mentions PRISMA methodology. Please include in the materials and methods section a brief description of this methodology and, also, at least one bibliographical reference.
Also, conclusions are really poor. Please enlarge them.
Author Response
The article entitled “Artificial intelligence and bone metastasis: a systematic review” is well-written and, from my point of view, would be of interest for the readers of Cancers. In spite of this and before its publication, I would like to make the authors the following suggestions:
Introduction section: a summary of the aim of the manuscript and, also, a description of the layout of the manuscript should be introduced.
- Answer: Thank you for the comment. We have explained the aim of the study in more detail.
Lines 86-88 please rewrite the sentence in order to make it easier to understand.
- Answer: Thank you for the comment. We rewrote the sentence.
Figure 1: mentions PRISMA methodology. Please include in the materials and methods section a brief description of this methodology and, also, at least one bibliographical reference.
- Answer: Thank you for the appropriate comment. We included in methods a description of PRISMA 2020 statement and reported an associated reference.
Also, conclusions are really poor. Please enlarge them.
- Answer: Thank you for the comment. We enlarged the conclusions as suggested.
Reviewer 2 Report
Comments and Suggestions for Authors
Need improvements

Comments on the Quality of English LanguageAuthor Response
The paper "Artificial Intelligence and Bone Metastasis: A Systematic Review" reviews 59 studies
on the use of AI in diagnosing and managing bone metastases. It highlights AI's effectiveness in
improving diagnostic accuracy through imaging techniques like bone scintigraphy and PET-CT
and underscores the potential for AI to enhance patient care. However, the manuscript need main
improvements:
1) The abstract lacks open questions and limitations of the existing survey that this survey
presented.
Answer: Thank you for the suggestion. We implemented the abstract by including open questions and limitations of the survey this study analyzed.
2) Lack of figures to give better representation.
Answer: Thank you. We inserted a figure showing “AI-based CT segmentation model for identification of bone metastatic lesion” as supplementary file.
3) There is no challenges section. Please add good example of deep learning methods in
medical images: Umirzakova, S., Mardieva, S., Muksimova, S., Ahmad, S. and Whangbo,
T., 2023. Enhancing the Super-Resolution of Medical Images: Introducing the Deep
Residual Feature Distillation Channel Attention Network for Optimized Performance and
Efficiency. Bioengineering, 10(11), p.1332.
Answer: Thank you for the appropriate comment. We inserted a challenges section in the discussion, and we mentioned the manuscript you indicated.
4) There is no metrics information are mainly used in bone metastasis cases. Kindly include
Answer: Thank you for the comment. We included further metrics information in the results.
5) Conclusion is not correlating with abstract and too short please include more information.
Rewrite.
Answer: Thank you for the comment. We enlarged the conclusions as suggested.
6) Discuss about the standard dataset available for bone metastasis cases.
Answer: Thank you for the appropriate comment. We reported the standard dataset used for bone metastasis in the results.
7) Kindly provide subjective analysis for different applications of bone metastasis cases.
Answer: Thank you for the appropriate comment. We provide further analysis for AI applications in bone metastasis.
8) Add one table of LIST OF IMPORTANT ACRONYMS used in paper.
Answer. Thank you for the comment. We added a list of acronyms as supplementary file.
9) Clearly state the objectives and scope of the survey paper, including the specific
methodologies and techniques covered.
Answer: Thank you for the comment. We have explained the aim of the study in more detail.
10) Consider adding a brief overview of the challenges and limitations of existing bone
metastasis methods.
Answer: Thank you for the appropriate comment. We reported challenges and limitations of existing methods for bone metastasis as you suggested.
11) Consider providing subheadings within each section to enhance readability and aid in
navigation. Use clear and descriptive section titles that accurately reflect the content of
each section.
Answer: Thank you for the appropriate comment. We have inserted subheadings within each section.
12) Consider adding more explanations and clarifications where necessary to ensure a
comprehensive understanding of the surveyed techniques.
Answer: Thank you. We added further explanations regarding the surveyed techniques.
13) Consider discussing potential future directions and open research challenges in bone
metastasis
Answer: Thank you for the insightful suggestion. We provided a detailed section on potential future directions and open research challenges in bone metastasis as you suggested.
Reviewer 3 Report
Comments and Suggestions for Authors
This work discusses the advancements in applying AI to bone metastasis. I have the following concerns regarding this work:
- Title: The title could be more specific, mentioning whether the AI application is aimed at aiding the treatment or diagnosis of bone metastasis.
- Purpose: It is not clear why the authors conducted this systematic review. Typically, a review addresses a particular problem or issue and aims to find a solution. It is not sufficient to conduct a review simply to summarize existing papers.
- Introduction: The authors might consider first discussing the limitations of AI in cancer treatment and diagnosis in the Introduction section. Including significant references, such as Siddique et al. (Rep Pract Oncol Radiother 2020; 25:656) and Chow et al. (Artificial Intelligence in Radiotherapy and Patient Care. In Artificial Intelligence in Medicine. Lidströmer N., Ashrafian H. (eds). Springer, Cham, Chapter 91, pp.1276-1285, 2022. https://doi.org/10.1007/978-3-030-58080-3_143-1), would strengthen this section.
- Figures and Tables: Figure 1, the PRISMA 2020 flow diagram, appears outdated. Additionally, Table 1 includes publications only up to 2023, with no papers from the current year.
- Future Prospects: The authors should provide a detailed section on the future prospects of AI applications in bone metastasis.
- References: There are only 86 references in this review. Typically, a systematic review should contain over 100 references.
No problem to read this paper.
Author Response
This work discusses the advancements in applying AI to bone metastasis. I have the following concerns regarding this work:
- Title: The title could be more specific, mentioning whether the AI application is aimed at aiding the treatment or diagnosis of bone metastasis.
Answer: Thank you for the appropriate comment. We edited the title following your suggestion.
- Purpose: It is not clear why the authors conducted this systematic review. Typically, a review addresses a particular problem or issue and aims to find a solution. It is not sufficient to conduct a review simply to summarize existing papers.
Answer: Thank you for the comment. We have explained the aim of the study in more detail.
- Introduction: The authors might consider first discussing the limitations of AI in cancer treatment and diagnosis in the Introduction section. Including significant references, such as Siddique et al. (Rep Pract Oncol Radiother 2020; 25:656) and Chow et al. (Artificial Intelligence in Radiotherapy and Patient Care. In Artificial Intelligence in Medicine. Lidströmer N., Ashrafian H. (eds). Springer, Cham, Chapter 91, pp.1276-1285, 2022. https://doi.org/10.1007/978-3-030-58080-3_143-1), would strengthen this section.
Answer: Thank you for the appropriate comment. We improved the introduction section as you suggested, and we mentioned the manuscripts you indicated.
- Figures and Tables: Figure 1, the PRISMA 2020 flow diagram, appears outdated. Additionally, Table 1 includes publications only up to 2023, with no papers from the current year.
Answer: Thank you for your comment. However, we used “PRISMA 2020 flow diagram for new systematic reviews which included searches of databases and registers only”, which is the last version of PRISMA 2020 statement. Moreover, we included in methods a description of PRISMA 2020 statement and reported an associated reference.
Finally, the electronic search on online databases was conducted on 31 December 2023. Therefore, unfortunately, we have not included papers from the current year.
- Future Prospects: The authors should provide a detailed section on the future prospects of AI applications in bone metastasis.
Answer: Thank you for the insightful suggestion. We provided a detailed section on the future prospects of AI applications in bone metastasis as you suggested.
- References: There are only 86 references in this review. Typically, a systematic review should contain over 100 references.
Answer: Thank you for the comment. We increased the number of the references.
Reviewer 4 Report
Comments and Suggestions for Authors
1. There is no review of AI theory. It is purely a classification and statistics of bone metastases medical methods.
2. There are too many tables, making it difficult for readers to read. This manuscript is a technical work report and does not make much academic contribution.
3. AI has many methods, and analysis and academic discussion should be conducted on how different AIs correspond to different medical behaviors.
4. The theory is insufficient and the academic contribution is weak. It is not recommended to be published.
Comments on the Quality of English LanguageThere are many grammatical errors in the present continuous tense in English. Not every word needs to be added with 'ing'.
Author Response
REVIEWER 4
- There is no review of AI theory. It is purely a classification and statistics of bone metastases medical methods.
- There are too many tables, making it difficult for readers to read. This manuscript is a technical work report and does not make much academic contribution.
- AI has many methods, and analysis and academic discussion should be conducted on how different AIs correspond to different medical behaviors.
- The theory is insufficient and the academic contribution is weak. It is not recommended to be published.
Thank you for your comments. We are sorry that you did not appreciate our work. However, we have modified the manuscript, focusing more on the effectiveness of AI in detection, management and prognosis of bone metastasis. Therefore, we hope you can appreciate the new version of the paper.
Round 2
Reviewer 2 Report
Comments and Suggestions for Authors
I give full support for publishing.
Reviewer 3 Report
Comments and Suggestions for Authors After reviewing it with the authors' responses, I am satisfied with the modifications and additional contents added by the authors as per my comments. They have addressed all my concerns. I can now recommend this work for publicationComments on the Quality of English Language
No comment.
Reviewer 4 Report
Comments and Suggestions for Authors
I think it can be accepted for publiction.